# Application of HPLC Fingerprint Combined with Chemical Pattern Recognition and Multi-Component Determination in Quality Evaluation of *Echinacea purpurea* (L.) Moench

**DOI:** 10.3390/molecules27196463

**Published:** 2022-09-30

**Authors:** Xuzhen Lv, Shuai Feng, Jiacheng Zhang, Sihai Sun, Yannan Geng, Min Yang, Yali Liu, Lu Qin, Tianlun Zhao, Chenxi Wang, Guangxu Liu, Feng Li

**Affiliations:** 1College of Pharmacy, Shandong University of Traditional Chinese Medicine, Jinan 250355, China; 2Department of Cardiology, Affiliated Hospital of Shandong University of Traditional Chinese Medicine, Jinan 250014, China; 3Department of Pharmacy, Liaocheng People’s Hospital, Liaocheng 252000, China; 4Department of Pharmacy, Affiliated Hospital of Shandong University of Traditional Chinese Medicine, Jinan 250014, China

**Keywords:** *Echinacea purpurea* (L.) Moench, high-performance liquid chromatography, fingerprint, chemical pattern recognition, hierarchical cluster analysis, principal component analysis

## Abstract

*Echinacea purpurea* (*EP*) is a common medicinal material for extracting anti-RSV components. However, up to now, there has been no effective and simple method to comprehensively reflect the quality of *EP*. In our current study, the quality of *Echinacea purpurea* (L.) Moench samples from six different cultivation locations in China was evaluated by establishing a high-performance liquid chromatography (HPLC) fingerprint, combining chemical pattern recognition and multi-component determination. In this study, the chemical fingerprints of 15 common peaks were obtained using the similarity evaluation system of the chromatographic fingerprints of traditional Chinese medicine (2012A Edition). Among the 15 components, three phenolic acids (caftaric acid, chlorogenic acid and cichoric acid) were identified and determined. The similarity of fingerprints of 16 batches of *Echinacea purpurea* (L.) Moench samples ranged from 0.905 to 0.998. The similarity between fingerprints of five batches of commercially available *Echinacea pupurea* (L.) Moench and the standard fingerprint ”R” ranged from 0.980 to 0.997, which proved the successful establishment of the fingerprint. PCA and HCA were performed with the relative peak areas of 15 common peaks (peak 3 as the reference peak) as variables. Anhui and Shaanxi can be successfully distinguished from the other four cultivation areas. In addition, the index components of caftaric acid, chlorogenic acid and cichoric acid were in the range of 1.77–8.60 mg/g, 0.02–0.20 mg/g and 2.27–15.87 mg/g. The results of multi-component index content determination show that the contents of the Shandong cultivation area were higher, followed by Gansu, Henan and Hebei, and the lowest were Anhui and Shaanxi. The results are consistent with PCA and HCA, which proved that the quality of *Echinacea purpurea* (L.) Moench from different origins was different. HPLC fingerprint combined with chemical pattern recognition and multi-component content determination was a reliable, comprehensive and prospective method for evaluating the quality of *Echinacea purpurea* (L.) Moench. This method provides a scientific basis for the quality control and evaluation of *Echinacea purpurea* (L.) Moench.

## 1. Introduction

*Echinacea purpurea* (L.) Moench is a perennial herb in the genus *Echinacea* of the Asteraceae family, which is native to North America [1]. It has been successfully introduced and is growing in many places in China, such as Beijing, Shanghai and Shandong. *Echinacea purpurea* (L.) Moench contains a variety of phenolic acids, such as caftaric acid, chlorogenic acid and cichoric acid (Figure 1), which have immunomodulatory, anti-inflammatory, antiviral and antitumor activities [2,3,4,5,6]. Pharmacological studies have shown that phenolic acids have antioxidant, free radical scavenging, antibacterial, antiviral, anti-inflammatory, immune boosting, hypoglycemic, hepatoprotective, uric acid-lowering and anti-atherosclerotic effects [7,8,9,10]. For the first time, a study documented that cichoric acid had an obvious inhibitory effect on RSV [11]. The uncertainty of the quality of medicinal materials can easily lead to the instability of their curative effect, so it is very important to control. At present, many studies have been conducted to evaluate the quality of *Echinacea purpurea* (L.) Moench, but a single quality control index cannot easily systematically evaluate the differences in the quality of *Echinacea purpurea* (L.) Moench from different origins. Therefore, in the present study, a high-performance liquid chromatography (HPLC) fingerprint of *Echinacea purpurea* (L.) Moench was established for the first time. The contents of caftaric acid, chlorogenic acid and cichoric acid in the aerial parts of *Echinacea purpurea* (L.) Moench were determined by the HPLC method. A principal component analysis (PCA) and hierarchical cluster analysis (HCA) were carried out by using the Origin software; three phenolic acids were identified, and their contents were determined [12,13]. The quality of *Echinacea purpurea* (L.) Moench from different origins was evaluated using this established method to provide a reference for the comprehensive quality evaluation of this medicinal material.

## 2. Results

### 2.1. Method Validation for Fingerprint Analysis

In the experimental study of precision and stability, the similarity of chromatograms is high, and the similarity between chromatograms is 1.000. In the repeatability experiment, the similarity of six samples was 0.999, 0.996, 0.999, 0.997, 1.000 and 0.998. The relative retention time and relative peak area of the common peak were investigated. The results show that the RSD was <5%. This shows that the method has good precision and stability, and the sample solution is stable without precipitation at room temperature.

### 2.2. Method Validation for Quantitative Analysis

After drawing the standard curve of the average peak area and concentration of three phenolic acids, a linear regression analysis was conducted. It was found that the linear relationship of all analytical components was good (r > 0.9993). The regression equation, linear range, correlation coefficient, quantitative limit and detection limit are shown in Table 1.

After the investigation of precision, the relative standard deviations of the three phenolic acids were 0.08%, 2.11% and 0.46%, respectively. The stability was good, and the RSD (%) was 0.64%, 2.57% and 1.91% of the peak area within 24 h. The repeatability was acceptable, and the RSD (%) was 0.19%, 0.83% and 0.61%, respectively. The accuracy of the method was verified by the sample recovery test. The average recovery of caftaric acid was 100.98%, and RSD (%) was 0.40%; chlorogenic acid was 98.06% and RSD (%) was 2.92%; cichoric acid was 99.41% and RSD (%) was 1.18%.

The above data show that the method used in the experiment is stable, reliable and of research significance.

### 2.3. HPLC Fingerprint Establishment

#### 2.3.1. Identification of Common Peaks

A total of 16 batches of *Echinacea pupurea* (L.) Moench powder were taken and prepared into sample solutions according to the method under “item 4.3.” Then, the sample solutions were injected into the HPLC and analyzed according to chromatographic conditions under “item 4.3.” The HPLC fingerprints of 16 batches of *Echinacea pupurea* (L.) Moench were obtained and imported into the Similarity Evaluation System for Chromatographic Fingerprint of Traditional Chinese Medicine (Version 2012A). S4 was selected as the reference fingerprint, the median method was used, and the time window width was 0.5. The chromatographic peaks with a separation degree greater than 1.5 were selected by multipoint calibration and matched automatically to generate superimposed fingerprints of *Echinacea pupurea* (L.) Moench (Figure 2) and reference fingerprint “R” (Figure 3A). Peak 3 was selected as the reference peak (S), because it was common in all the test samples and had good separation. Characteristic peaks were labeled with consecutive numbers (1, 2, 3, …, N). A total of 15 characteristic peaks were labeled, and the total area of these 15 characteristic peaks accounted for more than 92% of the total peak area. Therefore, these 15 peaks were identified as the common peaks in the HPLC fingerprints of *Echinacea pupurea* (L.) Moench.

By comparing their retention times with those of reference standards, three chromatographic peaks were identified, which were identified as caftaric acid (peak 3), chlorogenic acid (peak 4) and cichoric acid (peak 10). The HPLC reference fingerprints of samples ”R” and mixed reference standards are shown in Figure 3A,B.

#### 2.3.2. Similarity Analysis of the HPLC Fingerprints

HPLC chromatograms of the 16 batches of *Echinacea pupurea* (L.) Moench were imported into the Chinese Medicine Chromatographic Fingerprint Similarity Evaluation System (2012A Edition). After chromatographic peak matching, the standard fingerprint chromatogram”R” was generated, and the similarity between the fingerprints of different batches of samples and the standard fingerprint was calculated. The similarity between the fingerprints of 16 batches of *Echinacea pupurea* (L.) Moench samples and the standard fingerprint was 0.996, 0.967,0.993, 0.990, 0.995, 0.905, 0.977, 0.993, 0.993, 0.996, 0.994, 0.998, 0.998, 0.945, 0.994 and 0.995, respectively, indicating that the similarity of the 16 batches of *Echinacea pupurea* (L.) Moench samples was good.

#### 2.3.3. Verification of HPLC Fingerprints of Five Batches of Commercially Available *Echinacea pupurea* (L.) Moench from Different Origins

Five batches of commercially available *Echinacea pupurea* (L.) Moench powder (Y1–Y5, 2.500 g) were accurately weighed and prepared into sample solutions according to the method described in “item 4.3.” The sample solutions were then injected in the HPLC and analyzed under the chromatographic conditions described in “item 4.3.” The chromatograms were recorded. The HPLC chromatograms of five batches of *Echinacea pupurea* (L.) Moench and the reference fingerprint chromatogram “R” were imported into the Chinese Medicine Chromatographic Fingerprint Similarity Evaluation System (version 2012A). With the reference fingerprint chromatogram “R” as the reference peak, the standard fingerprint chromatogram (new) was generated after matching the chromatographic peaks, and its similarity was calculated. The results from the similarity analysis are shown in Table 2. The similarity between the fingerprints of five batches of commercially available *Echinacea pupurea* (L.) Moench and the standard fingerprint “R” was in the range of 0.980 to 0.997, which proved that the HPLC fingerprint of *Echinacea pupurea* (L.) Moench was successfully established.

### 2.4. Chemical Pattern Recognition

#### 2.4.1. Hierarchical Cluster Analysis (HCA)

The peak area of each common peak of *Echinacea pupurea* (L.) Moench was quantified relative to the peak area of the reference peak (peak 3, caftaric acid), which was then imported into Origin Pro software to obtain a 15 × 16-order matrix (Table 3). HCA was performed using Ward’s algorithm as the linkage method and the squared Euclidean distance as the measurement. The results show that 16 batches of samples were grouped into five clusters. Cluster I included SD(S1, S8, S9), HB(S13, S15); cluster II included GS(S3, S5), SD(S10, S11, S12), AH(S16); cluster III included SX(S4, S7); cluster IV included AH(S6, S14); and cluster V included SX(S2). The clustered heat map is shown in Figure 4.

#### 2.4.2. Principal Component Analysis (PCA)

To further explore the differences in the chemical components of *Echinacea pupurea* (L.) Moench samples from different regions of China, the peak area of each common peak of *Echinacea pupurea* (L.) Moench samples was quantified relative to the peak area of reference peak 3 (caftaric acid), which was then imported into Origin Pro software to obtain a 15 × 16-order matrix. PCA, an unsupervised pattern recognition method, was used to observe the natural aggregation of 16 batches of samples of *Echinacea pupurea* (L.) Moench from different origins. The first four principal components (PC1, PC2, PC3, PC4) with eigenvalues greater than 1 were selected. Their contribution rates were 38.08%, 24.01%, 11.66% and 10.43%, respectively. The cumulative contribution rate of the first four PCs reached 84.17%, indicating the good fitting ability of the model. It can be seen from the factor load results of the principal components (Table 4) that the information of principal component 1 mainly derived from peaks 4 (chlorogenic acid), 6, 8~9, 13 and 15; the information of principal component 2 mainly derived from peaks 1, 10 (cichoric acid) ~11; the information of principal component 3 mainly derived from peaks 2~3 (caftaric acid) and 12; and the information of principal component 4 mainly derived from peaks 5, 7 and 14.

The PCA biplot (loading and score plot) of the 16 batches of samples is displayed in Figure 5. It can be seen that the samples from different regions of China were well distinguished, especially the samples from Anhui (S6, S14) and Shaanxi (S2, S4, S7), which were clearly different from other regions. From the distance between the variable (common peaks) and the sample, the content of components in the seventh and 12th peaks of the Anhui cultivation area was higher; the content of components in the first, tenth (cichoric acid) and eleventh peaks was higher in the Shandong cultivation area; for the Shaanxi cultivation area, the content of components in the fifth peak of S4 and S7 was higher, while the content of components in the 15th peak of S2 was higher, which is consistent with the clustering result of S2 clustering as a single category.

#### 2.4.3. Determination of the Contents of Three Phenolic Acids in *Echinacea pupurea* (L.) Moench Samples

In total, 16 batches of samples of *Echinacea pupurea* (L.) Moench were taken and prepared into sample solutions according to the method described in “item 4.3.” After the samples solution was injected into the HPLC and analyzed under the chromatographic conditions, as described in “item 4.3”, the peak area was recorded, and the contents of the three phenolic acids were calculated according to the regression equation (Table 5). The cumulative summation graph of contents of the three components in all samples (Figure 6) was drawn with GraphPad Prism 7.0 software. It can be seen that the contents of these three components in samples of *Echinacea pupurea* (L.) Moench from different regions of China were quite different. In the 16 batches of samples, the contents of three index phenolic acid components were evaluated by region. The results show that the contents of the Shandong cultivation area were higher, followed by Gansu, Henan and Hebei, and the lowest were Anhui (S6, S14) and Shaanxi (S2, S4, S7). The results are consistent with those of HCA and PCA.

## 3. Discussion

In order to make the peak shape of the fingerprint attractive and informative, the effects of different extraction solvents (methanol, ethanol, 20% methanol, 40% methanol, 60% methanol and 80% methanol) on the peak shape and peak area of chromatographic peaks were compared, and it was concluded that 40% methanol was the most effective extraction solvent. In addition, to balance the baseline, the flow rate was selected to decrease to 0.8 mL/min at 25 min. The chromatographic peak shapes and resolution at different wavelengths were compared by full wavelength scanning. The results show that the best detection wavelength is 330 nm. In addition, the column temperature, the injection amount and the pH value of formic acid aqueous solution in the mobile phase were also investigated. Finally, the gradient elution conditions were: column temperature: 35 °C; injection volume: 6 μL; mobile phase: 0.7% formic acid aqueous solution acetonitrile. See “item 4.3” for gradient elution procedure.

In this experiment, the HPLC fingerprints of 16 batches of *Echinacea pupurea* (L.) Moench samples were established, and 15 common peaks were summarized, among which three, four and 10 were caftaric acid, chlorogenic acid and cichoric acid, respectively. The similarity evaluation result of the 16 batches was above 0.9, and five batches of commercially available Echinacea purpurea were used to verify the similarity with the reference fingerprint “R”. The result was 0.980 to 0.997, which proved that the fingerprint of *Echinacea pupurea* (L.) Moench was successfully established [14]. By pattern recognition (HCA and PCA) [15,16], Anhui and Shaanxi can be separated from the other four planting areas. Moreover, the content of peak 15 in S2 was found to be high by PCA, so Shaanxi samples S2, S4 and S7 were divided into two categories in the HCA. By measuring the contents of caftaric acid, chlorogenic acid and cichoric acid in the samples, the results show that the contents of the three components in the samples from different regions were quite different. The results show that the content was higher in Shandong, followed by Gansu, Henan and Hebei, and the lowest was in Anhui and Shaanxi. It is necessary to further study and explore the content differences of the three components in *Echinacea pupurea* (L.) Moench samples from different habitats and the reasons for their instability. The quality control and fine variety breeding of *Echinacea pupurea* (L.) Moench should be further strengthened.

## 4. Materials and Methods

### 4.1. Plant Materials, Chemicals, and Reagents

Different batches of *Echinacea purpurea* (L.) Moench were collected from the Hebei, Shandong, Shaanxi, Gansu, Henan and Anhui provinces of China in September 2021. A total of 5 batches of *Echinacea* were purchased from the Internet. The samples were authenticated as the dry aerial parts of *Echinacea pupurea* (L.) Moench in the genus *Echinacea* of the Asteraceae family by Prof. Li Feng from the Department of Traditional Chinese Medicine authentication at Shandong University of Traditional Chinese Medicine. The details of *Echinacea purpurea* (L.) Moench samples from different origins are listed in the Table 6. Voucher specimens were deposited at the Herbarium in the Shandong University of Traditional Chinese Medicine.

Caftaric acid (C_13_H_12_O_9_, P29J12F13966), chlorogenic acid (C_16_H_18_O_9_, Y20A11K11541) and cichoric acid (C_22_H_18_O_12_, Y02J12Y136139) were purchased from Shanghai Yuanye Biotechnology Co., Ltd. (Shanghai, China). The purity was more than 98%. In addition, analytical grade formic acid (210221) and methanol (210759) were purchased from Tianjin comeo Chemical Reagent Co., Ltd. (Tianjin, China), and HPLC-grade acetonitrile (AS1122-801) was purchased from Fisher Scientific Company (Waltham, MA, USA). Wahaha Purified water was provided by Hangzhou Wahaha Group Co., Ltd. (Hangzhou, China).

### 4.2. Instruments

The Agilent 1260 high performance liquid chromatograph system (Agilent Technologies, Waldbronn, Germany) was used, which is composed of a G1311C four-way pump, degasser, automatic sampler, G1314F variable wavelength UV detector and G1316A column temperature box. M8307AA chromatographic workstation software was used for system control and data analysis; electronic balance (FA2004); numerical control ultrasonic cleaner (KQ-500DB, Kunshan City Ultrasonic Instrument Co., Ltd., Kunshan, China); low-temperature circulating vacuum pump (DLSB-ZC, Zhengzhou Greatwall Scientific Industrial and trade Co., Ltd., Zhengzhou, China).

### 4.3. HPLC Fingerprint Chromatogram of Echinacea purpurea (L.) Moench

Before preparing the sample solution, the dried *Echinacea* was broken into powder and sieved through 40 mesh. The powder (2.500 g) was accurately weighed and placed into an Erlenmeyer flask. After 50 mL of 40% methanol solution was added with a solid–liquid ratio of 1:20, the mixture was accurately weighed and stood for 30 min in the dark. Then the mixture was extracted once for 30 min by ultrasound and cooled to room temperature. Methanol (40%) was then added to make up for the weight loss of the solution and mixed well. The extracts were then filtered through a 0.22 μm organic membrane filter to obtain sample solutions.

The analysis was performed on a ZORBAX SB-C18 (250 mm × 4.6 mm, 5 μm) column. The mobile phase consisted of 0.7% formic acid aqueous solution (A) and acetonitrile (B). The injection volume was 6 μL. The column temperature was maintained at 35 °C. The diode array detection (DAD) wavelength was 330 nm. Gradient elution conditions are shown in Table 7.

### 4.4. Validation of the HPLC Fingerprints Method

The precision was continuously evaluated with the same batch of *Echinacea* solution (S4) six times. The same sample solution was used for stability tests at 0, 2, 4, 6, 8, 10, 12 and 24 h. In addition, six solutions were prepared in parallel with the same sample powder (S4) to measure repeatability. Peak 3 (caftaric acid) was selected as the reference peak, and the RSD of the relative retention time and relative peak area of each common peak was calculated.

### 4.5. Determination of Three Phenolic Acids in Echinacea purpurea (L.) Moench

#### 4.5.1. Preparation of Mixed Standard Solution and Sample Solution

The caftaric acid, chlorogenic acid and cichoric acid were accurately weighed and dissolved with 70% methanol to obtain mixed standard solutions with a concentration of 200.00 μg/mL, 40.00 μg/mL and 1000.00 μg/mL, respectively, for caftaric acid, chlorogenic acid and cichoric acid. The standard solutions were stored in a refrigerator at 4 °C for future use.

The preparation method of sample solution was consistent with “item 4.3.”

#### 4.5.2. Chromatographic Conditions

The chromatographic conditions for sample injection analysis were in accordance with “item 4.3.”

### 4.6. Validation of the Quantitative Method

A total of 40, 80, 160, 320, 640 and 1000 μL of mixed standard solutions were taken and place in a 1 mL volumetric flask. Methanol solution (70%) was then added to each volumetric flask. Subsequently, the standard solutions were injected into the HPLC and analyzed according to the chromatographic method. The calibration curves were plotted with the concentration (µg/mL) as the abscissa (*X*) and the mean peak area as the ordinate (*Y*). The detection limit was the concentration of a standard solution with a signal-to-noise ratio (S/N) of 3 (LOD), and the quantitative limit was the concentration of the standard solution with an S/N of 10 (LOQ), respectively.

The precision was continuously evaluated with the same batch of *Echinacea* solution (S4) six times. The same sample solution was used for stability tests at 0, 2, 4, 6, 8, 10, 12 and 24 h. In addition, six solutions were prepared in parallel with the same sample powder (S4) to measure repeatability. The RSD of the peak area of caftaric acid, chlorogenic acid and cichoric acid was used as the measurement index.

The recovery test is used to evaluate the accuracy of the index phenolic acid. Six aliquots of the same batch of *Echinacea* powder (S4, 1.250 g) were removed and accurately weighed. To this, precise amounts of caftaric acid, chlorogenic acid and cichoric acid were added, respectively. Then, the solution was prepared as described above and the chromatographic analysis was performed. Each sample was tested three times. The average recovery percentage was evaluated by calculating the ratio of the detected amount to the added amount.

### 4.7. Data Analysis

Peak calibration and similarity analysis were performed according to the software of “chromatographic fingerprint similarity evaluation system of traditional Chinese medicine” recommended by the China Pharmacopoeia Commission (China Pharmacopoeia Commission, 2012). The similarity was calculated by the vector angle cosine method. The software SPSS (SPSS USA, version 20.0) and OriginPro (2019b) were used for the principal component analysis (PCA)and hierarchical cluster analysis (HCA) to evaluate the quality of *Echinacea*.

## 5. Conclusions

In this study, the fingerprint of *Echinacea purpurea* (L.) Moench was successfully established, and the feasibility of the fingerprint was verified with five batches of Echinacea purpurea sold in the market. Secondly, PCA and HCA were used to identify *Echinacea purpurea* (L.) Moench in different habitats. Anhui and Shaanxi planting areas can be separated from the other four planting areas. In addition, caffeic acid, chlorogenic acid and cichoric acid with antioxidant, anti-inflammatory and antiviral effects were selected as the indicator phenolic acid components to further compare the quality of *Echinacea purpurea* (L.) Moench in different cultivation areas. Therefore, the method based on HPLC fingerprint combined with chemical pattern recognition and multi-component content determination can realize the comprehensive evaluation of the quality of *Echinacea purpurea* (L.) Moench from different habitats. This method provides a basis for the extraction and separation of cichoric acid from *Echinacea purpurea* (L.) Moench in the future, as well as provides a scientific basis for the quality control of *Echinacea purpurea* (L.) Moench.

## Figures and Tables

**Figure 1 molecules-27-06463-f001:**
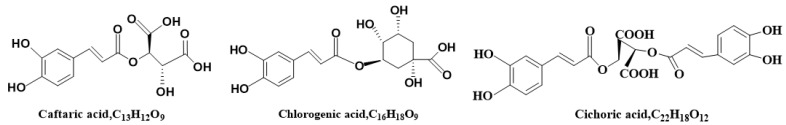
Chemical structure of representative components in *Echinacea purpurea* (L.) Moench.

**Figure 2 molecules-27-06463-f002:**
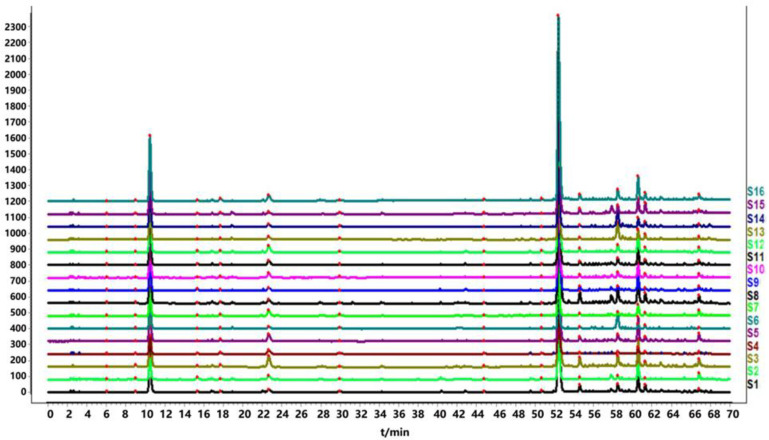
Superimposed fingerprints of 16 batches of *Echinacea purpurea* from different habitats.

**Figure 3 molecules-27-06463-f003:**
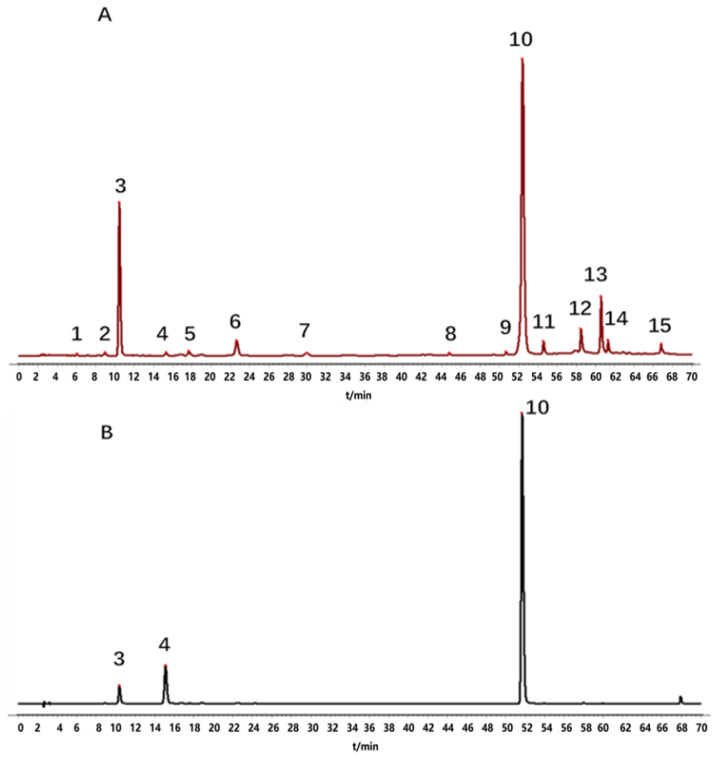
HPLC diagram of control fingerprint”R” (**A**) and mixed control solution (**B**) of *Echinacea purpurea* (L.) Moench: (3) caftaric acid, (4) chlorogenic acid, (10) Cichoric acid.

**Figure 4 molecules-27-06463-f004:**
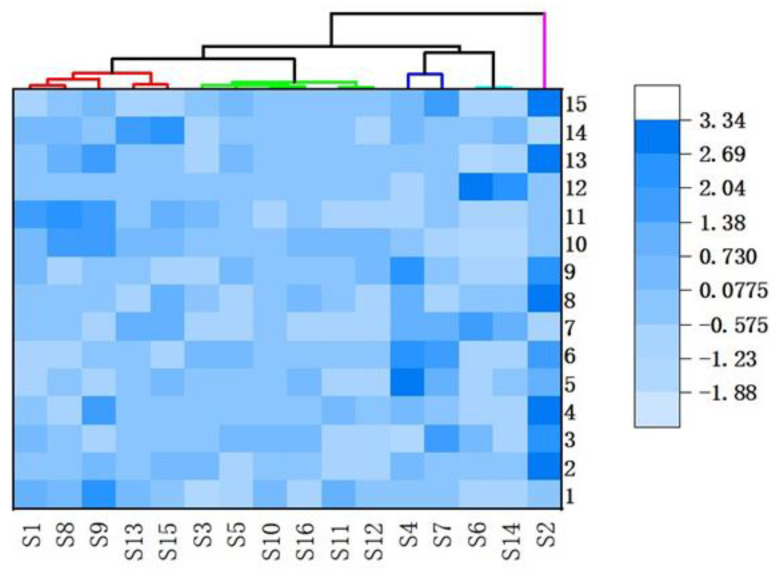
Cluster analysis heat map of *Echinacea purpurea* (L.) Moench.

**Figure 5 molecules-27-06463-f005:**
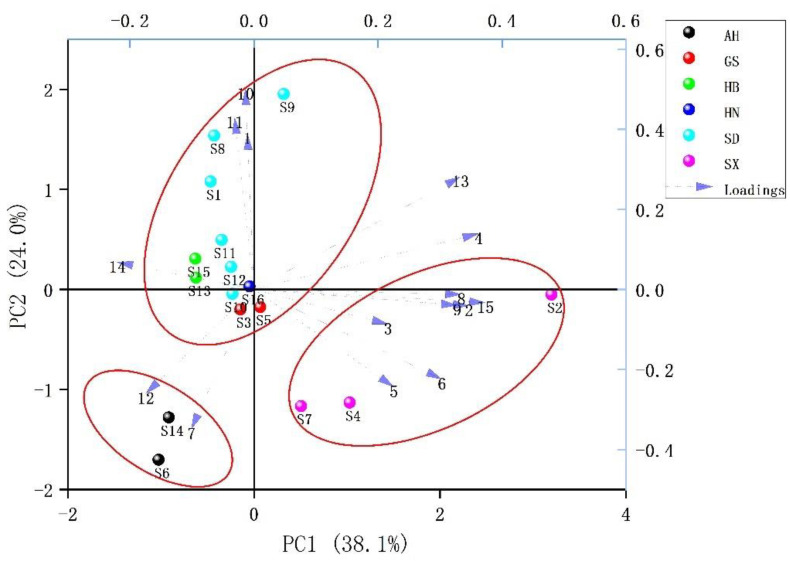
Principal component analysis biplot of *Echinacea purpurea* (L.) Moench.

**Figure 6 molecules-27-06463-f006:**
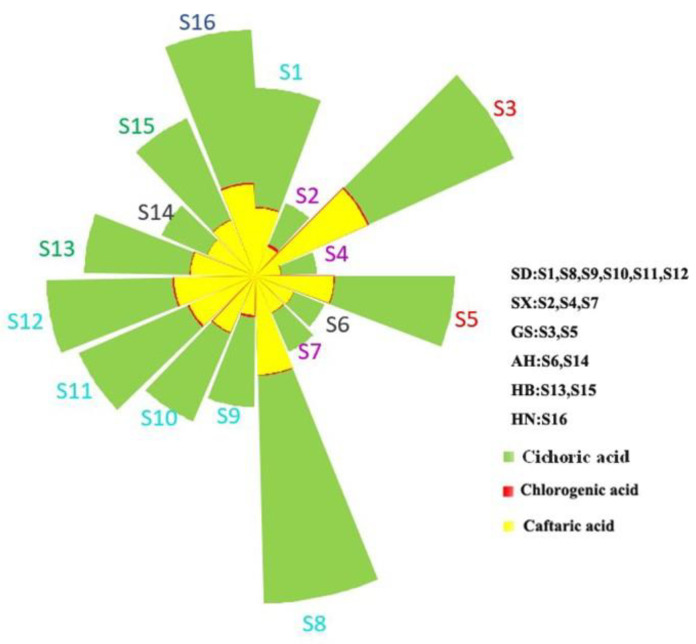
Cumulative sum of the contents of three phenolic acids in 16 batches of *Echinacea pupurea* (L.) Moench.

**Table 1 molecules-27-06463-t001:** Regression equation, correlation coefficient (R^2^), linear ranges, limit of quantitation (LOQ) and detection (LOD) of three phenolic acids.

Components	Regression Equation	R^2^	Linear Ranges (μg/mL)	LOD (μg/mL)	LOQ (μg/mL)
Caftaric acid	*Y* = 14.006*X* − 1.3253	0.9995	7.142~198.317	1.101	3.050
Chlorogenic acid	*Y* = 18.216*X* + 0.5184	0.9995	1.821~39.634	0.691	1.778
Cichoric acid	*Y* = 28.312*X* − 494.63	0.9997	41.867~994.947	17.909	19.113

**Table 2 molecules-27-06463-t002:** Similarity of fingerprints of 5 batches of commercially available *Echinacea pupurea* (L.) Moench samples from different origins.

Sample No.	Standard Fingerprint “R”	Y1	Y2	Y3	Y4	Y5	Standard Fingerprint (New)
Standard fingerprint “R”	1.000	0.977	0.989	0.996	0.988	0.979	0.997
Y1	0.977	1.000	0.986	0.976	0.985	0.958	0.984
Y2	0.989	0.986	1.000	0.991	0.993	0.972	0.995
Y3	0.996	0.976	0.991	1.000	0.993	0.98	0.998
Y4	0.988	0.985	0.993	0.993	1.000	0.965	0.993
Y5	0.979	0.958	0.972	0.98	0.965	1.000	0.986
Standard fingerprint (new)	0.997	0.984	0.995	0.998	0.993	0.986	1.000

**Table 3 molecules-27-06463-t003:** The quantification of the common peak area of *Echinacea pupurea* (L.) Moench 15 × 16-order matrix (peak 3 is the control peak).

Common Peak No.	Retention Time (Min)	S1	S2	S3	S4	S5	S6	S7	S8	S9	S10	S11	S12	S13	S14	S15	S16
1	6.061	0.012	0.007	0.003	0.008	0.004	0.005	0.006	0.010	0.018	0.009	0.011	0.008	0.010	0.005	0.007	0.005
2	8.967	0.020	0.053	0.023	0.027	0.013	0.021	0.022	0.017	0.026	0.018	0.013	0.013	0.020	0.021	0.024	0.022
3 (reference peak)	10.462	1.000	1.000	1.000	1.000	1.000	1.000	1.000	1.000	1.000	1.000	1.000	1.000	1.000	1.000	1.000	1.000
4	15.315	0.025	0.121	0.014	0.035	0.015	0.013	0.019	0.013	0.071	0.023	0.034	0.026	0.019	0.007	0.017	0.022
5	17.706	0.029	0.054	0.034	0.091	0.043	0.032	0.062	0.033	0.029	0.034	0.030	0.031	0.043	0.035	0.043	0.047
6	22.658	0.082	0.260	0.200	0.329	0.189	0.110	0.264	0.087	0.144	0.146	0.130	0.164	0.125	0.117	0.072	0.126
7	29.913	0.036	0.029	0.017	0.074	0.026	0.089	0.074	0.032	0.027	0.042	0.028	0.019	0.070	0.075	0.066	0.031
8	44.707	0.015	0.052	0.016	0.028	0.014	0.014	0.014	0.018	0.018	0.016	0.014	0.011	0.011	0.018	0.030	0.024
9	50.649	0.021	0.042	0.013	0.045	0.022	0.010	0.014	0.011	0.017	0.014	0.019	0.025	0.020	0.010	0.010	0.019
10	52.334	3.442	2.616	2.547	2.530	2.975	1.387	1.948	4.571	4.380	2.992	3.230	3.052	3.193	1.778	3.578	3.290
11	54.529	0.130	0.070	0.090	0.059	0.066	0.042	0.063	0.152	0.126	0.057	0.058	0.054	0.080	0.051	0.107	0.079
12	58.486	0.139	0.128	0.112	0.093	0.119	0.628	0.103	0.197	0.133	0.107	0.108	0.112	0.142	0.545	0.168	0.141
13	60.538	0.325	0.657	0.259	0.296	0.379	0.145	0.316	0.425	0.498	0.273	0.312	0.323	0.293	0.212	0.277	0.299
14	61.271	0.119	0.028	0.042	0.095	0.070	0.086	0.076	0.115	0.076	0.068	0.081	0.055	0.154	0.109	0.188	0.073
15	66.796	0.045	0.215	0.073	0.104	0.114	0.041	0.168	0.090	0.102	0.079	0.075	0.073	0.051	0.057	0.047	0.064

**Table 4 molecules-27-06463-t004:** Principal component factor load matrix of *Echinacea pupurea* (L.) Moench.

Peak No.	PC1	PC2	PC3	PC4
1	−0.011	0.389	0.031	0.224
2	0.346	−0.041	0.361	0.013
3	0.216	−0.087	0.374	−0.286
4	0.362	0.139	0.145	−0.065
5	0.225	−0.244	−0.137	0.507
6	0.303	−0.225	−0.334	0.154
7	−0.101	0.349	0.321	0.388
8	0.334	−0.012	0.324	0.085
9	0.328	−0.040	−0.213	0.270
10	−0.014	0.500	−0.032	0.159
11	−0.031	0.429	0.205	0.147
12	−0.175	−0.261	0.398	−0.163
13	0.333	0.281	0.079	−0.073
14	−0.221	0.068	0.340	0.511
15	0.373	−0.033	−0.021	−0.106

**Table 5 molecules-27-06463-t005:** Contents of 3 components in samples of *Echinacea pupurea* (L.) Moench from different origins (*n* = 3, mg/g).

Sample No.	Caftaric Acid (mg/g)	Chlorogenic Acid (mg/g)	Cichoric Acid (mg/g)
S1	4.688	0.091	8.259
S2	2.180	0.202	3.098
S3	8.592	0.093	11.103
S4	1.770	0.048	2.493
S5	5.481	0.064	8.343
S6	2.906	0.029	2.272
S7	2.793	0.039	2.969
S8	6.899	0.070	15.874
S9	2.757	0.151	6.250
S10	4.305	0.075	6.647
S11	4.999	0.131	8.264
S12	5.639	0.111	8.792
S13	4.479	0.066	7.350
S14	3.611	0.020	3.453
S15	4.184	0.054	7.682
S16	6.358	0.107	10.622

**Table 6 molecules-27-06463-t006:** Details of *Echinacea purpurea* (L.) Moench samples from different origins.

Sample No.	Chinese Name	Latin Name	Cultivation Area
S1	Zizhuiju	*Echinacea pupurea* (L.) Moench	Baicaoyuan Experimental Field A, Shandong University of Traditional Chinese Medicine, Changqing District, Jinan City, Shandong Province, China
S2	Zizhuiju	*Echinacea pupurea* (L.) Moench	Jinshan Village, Lantian County, Xi’an City, Shaanxi Province, China
S3	Zizhuiju	*Echinacea pupurea* (L.) Moench	Suzhou District, Jiuquan City, Gansu Province, China
S4	Zizhuiju	*Echinacea pupurea* (L.) Moench	Xiasanguantang Village, Chang’an District, Xi’an City, Shaanxi Province, China
S5	Zizhuiju	*Echinacea pupurea* (L.) Moench	Jingangwan, Suzhou District, Jiuquan City, Gansu Province, China
S6	Zizhuiju	*Echinacea pupurea* (L.) Moench)	Fengyang County, Chuzhou City, Anhui Province, China
S7	Zizhuiju	*Echinacea pupurea* (L.) Moench	Ma’e Town, Lintong District, Xi’an City, Shaanxi Province, China
S8	Zizhuiju	*Echinacea pupurea* (L.) Moench	Baicaoyuan Experimental Field B, Shandong University of Traditional Chinese Medicine, Changqing District, Jinan City, Shandong Province, China
S9	Zizhuiju	*Echinacea pupurea* (L.) Moench	Baicaoyuan Experimental Field C, Shandong University of Traditional Chinese Medicine, Changqing District, Jinan City, Shandong Province, China
S10	Zizhuiju	*Echinacea pupurea* (L.) Moench	Zhaosi Village, Qingzhou, Weifang City, Shandong Province, China
S11	Zizhuiju	*Echinacea pupurea* (L.) Moench	Gaoliu Town, Qingzhou, Weifang City, Shandong Province, China
S12	Zizhuiju	*Echinacea pupurea* (L.) Moench	Kangda Traditional Chinese Medicine cultivation Professional Cooperative, Qingzhou, Weifang City, Shandong Province, China
S13	Zizhuiju	*Echinacea pupurea* (L.) Moench	Lingshou County, Shijiazhuang City, Hebei Province, China
S14	Zizhuiju	*Echinacea pupurea* (L.) Moench	Qiaocheng District, Bozhou City, Anhui Province, China
S15	Zizhuiju	*Echinacea pupurea* (L.) Moench	Yifeng Road, Anguo City, Hebei Province
S16	Zizhuiju	*Echinacea pupurea* (L.) Moench	Zhengyang County, Zhumadian City, Henan Province, China
Y1	Zizhuiju	*Echinacea pupurea* (L.) Moench	Commercially available samples originating from Henan Province, China
Y2	Zizhuiju	*Echinacea pupurea* (L.) Moench	Commercially available samples originating from Tibet Autonomous Region, China
Y3	Zizhuiju	*Echinacea pupurea* (L.) Moench	Commercially available samples originating from Shanxi Province, China
Y4	Zizhuiju	*Echinacea pupurea* (L.) Moench	Commercially available samples originating from Hubei Province, China
Y5	Zizhuiju	*Echinacea pupurea* (L.) Moench	Commercially available samples originating from Yunnan Province, China

**Table 7 molecules-27-06463-t007:** Gradient elution conditions.

Time (min)	0.7% Formic Acid Aqueous Solution (A%)	Acetonitrile (B%)	Flow Rate (mL/min)
0	91.0	9.0	1.0
9	91.0	9.0	1.0
22	88.0	12.0	1.0
25	88.0	12.0	0.8
42	82.0	18.0	0.8
45	80.0	20.0	0.8
65	60.0	40.0	0.8
68	58.0	42.0	0.8
70	55.0	45.0	0.8

## Data Availability

Data is contained within the article. In additional, the data presented in this study are available on request from the corresponding author.

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
