# Peer review of "Application of HPLC Fingerprint Combined with Chemical Pattern Recognition and Multi-Component Determination in Quality Evaluation of Echinacea purpurea (L.) Moench"

_molecules, 2022, doi:10.3390/molecules27196463_

Round 1

Reviewer 1 Report

The chromatographic analysis (HPLC) of Echinacea purpurea identifying some secondary metabolites defined as fingerprints is presented in order to evaluate the quality of Echinacea purpurea samples grown in different regions of China. They validated the chromatographic method and compared the analysis results of 16 samples, demonstrating the efficiency and practicability of the method. 

1. What are the similarity criteria presented in Table 2. 

2. The first paragraph of the "discussion" section is descriptive; in this section an interpretation of the results is expected. 

Author Response

Dear Reviewers,

       Thank you for reading our manuscript and reviewing it, which will help us improve it to a better scientific level. We revised our manuscript, and quite a lot of changes have taken place. So we have sent the revised manuscript.

        Next, the key points mentioned by the reviewer will be discussed. Point 1: 1. What are the similarity criteria presented in Table 2.

        Response 1: The control fingerprint "R" obtained from 16 batches of Echinacea pupurea (L.) Moench samples is the reference fingerprint, which is imported into Similarity Evaluation System for Chromatographic Fingerprint of Traditional Chinese Medicine (Version 2012A) with 5 batches of commercially available Echinacea pupurea (L.) Moench HPLC fingerprints. After the fingerprints are superposed, the similarity between the fingerprints of 5 batches of commercially available samples and the standard fingerprint (newly generated) is calculated. In addition, Similarity Evaluation System for Chromatographic Fingerprint of Traditional Chinese Medicine (Version 2012A) uses the included angle cosine method to calculate the similarity between samples. Referring to the technical requirements of the fingerprint study of traditional Chinese medicine injections, the similarity calculation results are 0.9~1.0, which means that the samples are highly similar.

  1. The first paragraph of the "discussion" section is descriptive; in this section an interpretation of the results is expected.

Response 2: The discussion part has been revised, as shown in the revised draft, as follows:

  1. Discussion

In order to make the peak shape of the fingerprint good-looking and informative, the effects of different extraction solvents (methanol, ethanol, 20% methanol, 40% methanol, 60% methanol and 80% methanol) on the peak shape and peak area of chromatographic peaks were compared, and it was concluded that 40% methanol was the most effective extraction solvent. In addition, to balance the baseline, the flow rate was selected to decrease to 0.8 ml / min at 25 minutes. The chromatographic peak shapes and resolution at different wavelengths were compared by full wavelength scanning. The results show that the best detection wavelength is 330 nm. In addition, the column temperature, the injection amount and the pH value of formic acid aqueous solution in the mobile phase were also investigated. Finally, the gradient elution conditions were: column temperature: 35 ℃; Injection volume: 6 μ L;Mobile phase: 0.7% formic acid aqueous solution acetonitrile. See “item 4.3” for gradient elution procedure.

In this experiment, HPLC fingerprints of 16 batches of Echinacea pupurea (L.) Moench samples were established, and 15 common peaks were summarized, among which 3, 4 and 10 were caftaric acid, chlorogenic acid and cichoric acid respectively. The similarity evaluation result of 16 batches of samples was above 0.9, and 5 batches of commercially available Echinacea purpurea were used to verify the similarity with the reference fingerprint "R", the result was 0.980 to 0.997, which proved that the fingerprint of Echinacea pupurea (L.) Moench was successfully established. By pattern recognition (HCA and PCA), Anhui and Shaanxi can be separated from the other four planting areas. Moreover, the content of peak 15 in S2 was found to be high by PCA, so Shaanxi S2 samples, S4 and S7 were divided into two categories in the HCA. By measuring the contents of caftaric acid, chlorogenic acid and cichoric acid in the samples, the results showed that the contents of the three components in the samples from different regions were quite different. The results showed that the content was higher in Shandong, followed by Gansu, Henan and Hebei, and the lowest in Anhui and Shaanxi. In the future, it is necessary to further study and explore the content differences of the three components in Echinacea pupurea (L.) Moench samples from different habitats and the reasons for their instability. The quality control and fine variety breeding of Echinacea pupurea (L.) Moench should be further strengthened.

        Thank you for your careful review. We really appreciate your efforts in reviewing our manuscript during this unprecedented and challenging time. We give you our best wishes. Your careful review has helped to make our study clearer and more comprehensive.

Sincerely yours,

Xuzhen Lv

Reviewer 2 Report

The manuscript refers to study of the quality evaluation of Echinacea purpurea (L.) Moench samples from different geographical locations in China using HPLC fingerprint combined with chemometric methods. The experiment was well designed and performed. The Manuscript is mostly clearly written and adequately explaining the obtained results. However, some ambiguities exist and additional explanations are needed. I suggest publishing the paper after minor revision.

Some comments are included to improve the manuscript:

- spelling/lettering errors need to be corrected; spaces are missing throughout the text in many places

- Echinacea purpurea should be italicized throughout the Manuscript (lines: 136-137, 166, 168, 182, 336-337)

- the chicoric acid formula is not of acceptable quality

-the peak area of each common peak 16 of Echinacea pupurea (L.) Moench samples was quantified relative to the peak area of peak 3 (caftaric acid)… The table with these data should be added

- the loading plot is missing, and it is necessary for adequate interpretation of score plot, so it should be added

-the samples from Shaanxi (S2, S4, S7) were clearly different from other regions… Actually, sample S2 does not belong to the same cluster as S4 and S7 (according to PC score plot)

- the authors could add PC1/PC3 and PC2/PC3 score plots that maybe will provide additional information

- the conclusion is too extensive and some statements are repeated

Author Response

Dear Reviewers,

      Thank you for your valuable comments and suggestions. These comments are very valuable and helpful for revising and improving our paper. We have made corresponding modifications to the manuscript, and our reply point by point is given below.

        Point 1: spelling/lettering errors need to be corrected; spaces are missing throughout the text in many places.

        Response 1: We have carefully checked and corrected spelling errors and space errors according to the suggestions of the reviewer, and marked them in red in the manuscript.

  1. -Echinacea purpurea should be italicized throughout the Manuscript (lines: 136-137, 166, 168, 182, 336-337).

      Response 2: The names of Echinacea pupurea (L.) Moench appearing in the opposite lines: 136-137, 166, 168, 182, 336-337 were changed to italics, and the full text was checked and modified.

  1. the chicoric acid formula is not of acceptable quality.

     Response 3: The spelling of cichoric acid formula in the full text has been corrected, and its molecular structure diagram has been redrawn (in Figure 1). In addition, calibrate the batch number of cichoric acid standard.

  1. the peak area of each common peak 16 of Echinacea pupurea (L.) Moench samples was quantified relative to the peak area of peak 3 (caftaric acid)… The table with these data should be added.

         Response 4: About Echinacea pupurea (L.) Moench 15 × 16 The quantization matrix table of common peak area (peak 3 is the control peak) is increased. The added data shall be marked in red font in Table 3 of the revised version.

  1. the loading plot is missing, and it is necessary for adequate interpretation of score plot, so it should be added.

      Response 5: The load factor value is added to Table 4 in the revised version, and the load diagram is added to Figure 5 in the revised version. Through analysis, it can be seen from the factor load results of principal component (Table 4) that the in formation of principal component 1 mainly comes from lakes 4 (chlorine acid), 6, 8~9, 13 and 15; The information of principal component 2 mainly comes from peaks 1, 10 (cichoric acid)~11;  The information of principal component 3 mainly comes from peaks 2~3 (caftaric acid) and 12;  The information of principal component 4 mainly comes from peaks 5, 7 and 14.

  1. the samples from Shaanxi (S2, S4, S7) were clearly different from other regions… Actually, sample S2 does not belong to the same cluster as S4 and S7 (according to PC score plot).

       Response 6: By adding the load diagram (in Figure 5), it can be concluded that the distance between sample S2 and peak 15 in Shaanxi is relatively close, indicating that the content of components in peak 15 is relatively high, while the distance between S4 and S7 and peak 5 is relatively close, indicating that the content of components in peak 5 is relatively high. Therefore, Shaanxi sample S2 is clustered into one group in cluster analysis. The modification description is on lines 194-195.

  1. the authors could add PC1/PC3 and PC2/PC3 score plots that maybe will provide additional information.

        Response 7: Double plots (variable load plot and sample score plot) were added. From the distance between the variable (common peaks) and the sample, the content of components in the 7th and 12th peaks of Anhui cultivation area is higher;  The content of components in No. 1 and No. 10 (cichoric acid) No. 11 peaks was higher in Shandong cultivation area;  For Shaanxi cultivation area, the content of components in the 5th peak of S4 and S7 is higher, while the content of components in the 15th peak of S2 is higher, which is consistent with the clustering result of S2 clustering as a single category. The modification description is on lines 189-195.

  1. the conclusion is too extensive and some statements are repeated.

Response 8: The conclusion has been revised to make it concise and clear. Modify as follows:

  1. Conclusions

In this study, the fingerprint of Echinacea purpurea (L.) Moench was successfully established, and the feasibility of the fingerprint was verified with 5 batches of Echinacea purpurea sold in the market. Secondly, PCA and HCA were used to identify Echinacea purpurea (L.) Moench in different habitats. Anhui and Shaanxi planting areas can be separated from the other four planting areas. In addition, caffeic acid, chlorogenic acid and cichoric acid with antioxidant, anti-inflammatory and antiviral effects were selected as the indicator phenolic acid components to further compare the quality of Echinacea purpurea (L.) Moench in different cultivation areas. Therefore, the method based on HPLC fingerprint combined with chemical pattern recognition and mul-ti-component content determination can realize the comprehensive evaluation of the quality of Echinacea purpurea (L.) Moench from different habitats. This method provides a basis for the extraction and separation of cichoric acid from Echinacea purpurea (L.) Moench in the future, and also provides a scientific basis for the quality control of Echinacea purpurea (L.) Moench.

          We send our best wishes to you. Great to you for the time and effort you expend on this paper.

Sincerely yours,

Xuzhen Lv
